# Mouse Models with SGLT2 Mutations: Toward Understanding the Role of SGLT2 beyond Glucose Reabsorption

**DOI:** 10.3390/ijms24076278

**Published:** 2023-03-27

**Authors:** Keiko Unno, Kyoko Taguchi, Yoshiichi Takagi, Tadashi Hase, Shinichi Meguro, Yoriyuki Nakamura

**Affiliations:** 1Tea Science Center, University of Shizuoka, 52-1 Yada, Suruga-ku, Shizuoka 422-8526, Japan; gp1719@u-shizuoka-ken.ac.jp (K.T.); yori.naka222@u-shizuoka-ken.ac.jp (Y.N.); 2Production Center for Experimental Animals, Japan SLC, Inc., 85 Ohara, Kita-ku, Hamamatsu 433-8102, Japan; yoshiichi-takagi@jslc.co.jp; 3R & D, Kao Corporation, 2-1-3 Bunka, Sumida-ku, Tokyo 131-8501, Japan; hase.tadashi@kao.com; 4Biological Science Research, Kao Corporation, Akabane, Ichikai-machi, Haga-gun, Tochigi 321-3497, Japan; meguro.shinichi@kao.com

**Keywords:** amyloid beta (A4) precursor protein, glucose homeostasis, glucosuria, Jimbee mouse, SAMP10-ΔSglt2 mouse, SGLT2^−/−^ mouse, SGLT2 mutation, sodium–glucose cotransporter 2, Sweet Pee mouse

## Abstract

The sodium–glucose cotransporter 2 (SGLT2) mainly carries out glucose reabsorption in the kidney. Familial renal glycosuria, which is a mutation of SGLT2, is known to excrete glucose in the urine, but blood glucose levels are almost normal. Therefore, SGLT2 inhibitors are attracting attention as a new therapeutic drug for diabetes, which is increasing worldwide. In fact, SGLT2 inhibitors not only suppress hyperglycemia but also reduce renal, heart, and cardiovascular diseases. However, whether long-term SGLT2 inhibition is completely harmless requires further investigation. In this context, mice with mutations in SGLT2 have been generated and detailed studies are being conducted, e.g., the SGLT2^−/−^ mouse, Sweet Pee mouse, Jimbee mouse, and SAMP10-ΔSglt2 mouse. Biological changes associated with SGLT2 mutations have been reported in these model mice, suggesting that SGLT2 is not only responsible for sugar reabsorption but is also related to other functions, such as bone metabolism, longevity, and cognitive functions. In this review, we present the characteristics of these mutant mice. Moreover, because the relationship between diabetes and Alzheimer’s disease has been discussed, we examined the relationship between changes in glucose homeostasis and the amyloid precursor protein in SGLT2 mutant mice.

## 1. Introduction

The plasma glucose concentration is maintained within narrow limits (4–10 mmol/L), which prevents the morbid consequences of prolonged hypoglycemia and hyperglycemia. Glucose uptake by peripheral organs is essential to maintaining blood glucose levels within the normal range, and the regulation of glucose uptake in these tissues primarily involves insulin and glucagon. Insulin is secreted by pancreatic beta cells in response to elevated blood glucose levels and activates glucose transporters in various tissues to promote glucose uptake [1]. On the other hand, glucagon, which is also released from the pancreas, acts directly on the liver to stimulate glucose production and also decreases glucose uptake [2]. Disturbed glucose uptake is a feature of type 2 diabetes and metabolic syndrome.

In addition, the kidneys play an essential role in glucose homeostasis as they are involved in certain central processes, i.e., the filtration of glucose from large volumes of plasma via the glomerulus, gluconeogenesis in the proximal tubules, and reabsorption of glucose from the proximal tubule lumen [3]. Almost all of the sugar filtered by the kidneys is reabsorbed in the proximal tubules and is profoundly connected to the homeostasis of glucose metabolism.

Four types of glucose transporters are involved in glucose reabsorption in the human kidneys: sodium–glucose cotransporters (SGLT) 1 and 2 and glucose transporters (GLUT) 1 and 2 [4,5]. SGLT2 is predominantly expressed in the anterior half of the proximal tubule (the S1 and S2 segments) and has a low affinity and high transport capacity for glucose. It transports sodium and glucose in a 1:1 ratio. Approximately 90% of the sugar in the primary urine is reabsorbed by SGLT2, and the remaining 10% is transported by SGLT1, which has a high-affinity low-transport capacity in the posterior portion of the proximal tubule (S3 segment) [6]. GLUT 1 and 2 localize to the basolateral membrane of the proximal ruble of the kidney and are responsible for returning filtered glucose and newly produced glucose in the kidney to the circulation. GLUT1 has a high affinity for glucose and GLUT2 has a low affinity [6]. Among these transporters, mutations in SGLT2 are associated with familial renal glucosuria, in which glucose reabsorption from the urine is reduced without hyperglycemia or tubular dysfunction [7,8,9,10,11,12,13,14,15].

Much research has been conducted on the site of SGLT2 mutation, its genetic background, and its disease status, and new mutation sites have been discovered. In addition, several experimental animals with SGLT2 mutations have been established [16,17,18,19,20]. The development and study of experimental animals with SGLT2 mutations, and studies in humans, enhance our understanding of this genetic disease and the function of SGLT2. Furthermore, they provide valuable information for research into SGLT2 inhibitors. With the increasing number of diabetic patients worldwide [21,22], they are gaining attention as new therapeutic agents [23,24]. They are designed to improve diabetes by inhibiting the reabsorption of sugar from the kidneys and thereby improving hyperglycemia. The treatment of diabetic patients with SGLT2 inhibitors is reported to induce body weight reduction, decrease in HbA1c, and reduction in blood pressure [25]. In addition to lowering blood glucose levels, many concomitant effects have been reported, including reductions in kidney, heart, eye, and cardiovascular diseases [26,27,28,29,30,31,32,33]. Since the benefits of SGLT2 inhibitors in heart failure patients have been clearly demonstrated, SGLT2 inhibitors are now part of the guideline therapies for patients with heart failure [26,27]. SGLT2 inhibitors have also been identified as strong antioxidant medications [34]. On the other hand, the adverse effects of long-term inhibition of SGLT2 function requires careful monitoring [35,36,37,38].

Alzheimer’s disease (AD) is one of the most common dementias, and a high prevalence of impaired glucose tolerance or type 2 diabetes has been reported in AD patients [39]. Amyloid precursor protein (APP) is not only the source of amyloid-β (Aβ) peptides but also directly affects insulin and glucose regulation [40,41,42]. Although AD patients tend to have elevated blood glucose levels, glucose metabolism in the brain is decreased. As a reason for that, it has been explained that Aβ causes abnormal subcellular localization of the glucose transporter (GLUT1) in the brain parenchyma [43].

Herein, we present various examples of mice with SGLT2 mutations that have been reported in the literature, although the number of cases remains relatively small. Detailed examinations of the biological changes caused by SGLT2 mutations revealed that SGLT2 not only serves as a target for diabetes treatment through its function in glucose reabsorption in the kidney but is also associated with bone metabolism, life span, and brain function. We also show that SGLT2 mutation is associated with the altered *APP* expression in the brain.

## 2. Renal Glucosuria in Humans

In general, glucosuria occurs when blood glucose levels rise to the maximum kidney reabsorption capacity and is the most common cause of hyperglycosemia is diabetes. Hyperglycemia increases the risk of vascular complications, kidney disease, neuropathy, and retinopathy [44].

There are patients with familial renal glucosuria that have mutations in the solute carrier family 5 member 2 (Slc5a2) gene encoding SGLT2 [7,8,9,10,11,12,13,14,15]. Mutations in SGLT2 are easy to detect because it causes sugar to be excreted in the urine despite the patients not being diabetic. Despite the fact that SGLT2 plays a major role in glucose reabsorption, hypoglycemia is unlikely to occur in patients with familial glycosuria or even experiments with maximum-dose SGLT2 inhibitors [45].

SGLT2 is composed of 672 amino acids. The gene comprises 14 exons (spanning 8 kb) [46]. The transporter contains 14 putative transmembrane-spanning domains. Among various mammalian species, there is a high degree of homology both at the nucleotide and amino acid levels [5]. Depending on the site of mutation, the effect on SGLT2 function varies. According to a report on 86 mutations in Slc5a2 that caused familial renal glucosuria in humans, the mutations are primarily missense (75.6%), frameshift (8.1%), splicing (5.8%), and nonsense (4.7%) mutations, and the most common mutation sites are exon 11 (18.6%), exon 8 (12.8%), and exon 4 (11.6%) [10]. Different mutations have been reported to cause different symptoms of glycosuria. Homozygous mutations and compound heterozygous mutations exhibit more severe glucosuria [10]. Moreover, not all individuals with similar or identical mutations exhibit the same degree of increased glucose excretion [10]. These findings suggest the influence of other genes in glucose transport, in addition to non-genetic factors such as diet and various environmental factors.

### 2.1. SGLT2 Mutation and SGLT2 Inhibitor

Renal glucosuria caused by mutations in SGLT2 is not generally considered to be associated with serious physical and clinical consequences [12]. A large cross-sectional study of renal glucose in adolescents reported that glucose reduction was associated with lower body weight and lower systolic blood pressure [13]. These data support the utility of SGLT2 inhibitors as a treatment for diabetes [47].

SGLT2 inhibitors are not completely selective for SGLT2 and inhibit SGLT1 to different degrees. However, they are commonly referred to as SGLT2 inhibitors because their affinity for SGLT2 is considerably higher than for SGLT1 [48]. Although SGLT2 inhibition raises concerns about the risk of hypoglycemia, previous studies reported that SGLT2 inhibition increased glucose loading to the late proximal tubules, enhanced the glucose transport capacity of SGLT1, and limited glucose excretion and the risk of hypoglycemia. SGLT2 inhibitors also increase plasma glucagon levels and promote gluconeogenesis in the liver. They also promote lipolysis and shift substrate utilization from carbohydrates to lipids [49]. These biological regulatory systems may make hypoglycemia less likely with SGLT2 inhibitors.

SGLT2 inhibitors have not only been found to lower blood glucose levels but are also associated with energy metabolism, mitochondrial function, oxidative stress, autophagy, apoptosis, and inflammation [48]. In addition, SGLT2 inhibitors have been shown to have potential roles in neuronal damage states, primarily Alzheimer’s disease and cerebral ischemia [50].

### 2.2. Other Kidney Glucose Transporters and Their Diseases

In glucose-galactose malabsorption disease, the inability of the small intestinal epithelium to absorb glucose, galactose, and their constituent molecules due to mutations in SGLT1 results in severe diarrhea at the onset of postnatal feeding. This is followed by severe dehydration, hypoglycemia, and acidosis, and, within a few days or weeks, the child is severely dehydrated, hypoglycemic, and acidotic [51].

Fanconi–Bickel syndrome, a rare syndrome originally named hepatorenal glycosis with renal Fanconi syndrome, is caused by mutations in GLUT2. It results in systemic solute transport dysfunction in the proximal tubules, causing the accumulation of glycogen and glucose in these cells [52]. GLUT1 is known to have an important role in vertebrate brain development [46]. GLUT1 deficiency syndrome is a neurological disorder caused by reduced expression and function of GLUT1 in the brain. In Alzheimer’s disease, GLUT1 expression is reduced in the cerebral cortex and hippocampus [53].

## 3. Renal Glucosuria Models in Experimental Animals

Several experimental animals with SGLT2 mutations have been reported in the literature. Three models were artificially induced [16,17,18,19] and one mutation occurred spontaneously [20]. SGLT2 mutant mice were developed to elucidate the role of SGLT2 in the kidney and in glucose homeostasis. They are also used to study the biological effects of long-term SGLT2 inhibitor treatment, which is attracting attention as a new diabetes treatment. Compared to wild-type mice, these mice exhibited glycosuria, polyuria, and increased food and water intake, but no difference in plasma glucose concentration [26,27,28,29]. However, one mouse line exhibited a slight but significant decrease in plasma glucose concentration [20].

### 3.1. SGLT2^−/−^ Mouse

SGLT2^−/−^ mice were generated by homologous recombination of the Slc5a2 gene with a targeting vector, replacing exons 1–5 with a selection cassette [16]. SGLT1 mRNA expression was reduced by approximately 40% in SGLT2^−/−^ compared to wild-type mice. Kidney GLUT1 and GLUT2 mRNA expression exhibited no significant differences between genotypes [16]. The mice exhibited no changes in body weight or urinary excretion of other proximal tubular substrates besides glucose. The blood glucose levels did not differ between the mutant and wild-type breeder pairs (132 ± 2 versus 134 ± 4 mg/dL) [16]. SGLT2^−/−^ mice exhibited increased food intake compared to wild-type mice, perhaps in response to urinary glucose and calorie losses.

It was found that inducing diabetes into SGLT2 knockout mice (C57BL/6 background) with low doses of streptozotocin attenuated hyperglycemia and glomerular hyperfiltration. These data supported the concept that the inhibition of SGLT2 increases glucose excretion into the urine and lowers blood glucose levels during diabetes [54]. SGLT2 knockout mice fed a high-fat diet were protected from hyperglycemia, had impaired glucose tolerance, and had reduced plasma insulin concentrations compared to normal mice [18]. SGLT2-deficient mice on a db/db background were protected from fasting hyperglycemia and had dramatically improved plasma insulin levels. In addition, pancreatic β-cell function was preserved [18]. These data support the efficacy of SGLT inhibition as an insulin-independent treatment for diabetes.

### 3.2. Sweet Pee Mouse

The Sweet Pee mutant was generated by N-ethyl-N-nitrosourea (ENU) mutagenesis on a C57BL6 background. The mutation was identified as a single thymine insertion at position 433 in exon 4, and this frameshift mutation resulted in a premature stop signal [17]. Sweet Pee mutant mice exhibited improved glucose tolerance, higher urinary excretion of calcium and magnesium, and growth retardation [17]. In Sweet Pee mice in which diabetes was induced with low-dose streptozotocin, despite improved glycemic control, a higher mortality rate was observed compared to diabetic wild-type mice [17]. Inhibition of SGLT2 has been reported to potentially improve glucose control but may increase the risk of infection, malnutrition, and mortality [17]. The mutant mice showed normal serum calcium and phosphate, parathyroid hormone, vitamins D, and fibroblast growth factor levels [55]. Although no changes in trabecular or cortical bone micro-structure were observed in Sweet Pee mice, a decrease in cortical bone mineral density was reported compared to wild-type mice at 25 weeks [55]. Loss of SGLT2 function in mice may contribute to long-term bone fragility.

### 3.3. Jimbee Mouse

The Jimbee mouse carries an ENU-induced 19-base-pair deletion on chromosome 7 within exon 10 of the Scl5a2 (*Sglt2*) gene, generated on a C57BL/6J background [19]. Jimbee mice exhibited glucose homeostasis. However, they developed chronic polyuria, glycosuria, and hypercalciuria [19]. Several components of this phenotype, such as growth retardation, were only observed in male mice [19]. Blood glucose concentrations depended on genotype, sex, and age, but there were no significant differences within each over time [19]. It was demonstrated that chronic blockage of SGLT2 reduced the mineralization of bone but not its fracture resistance in this model [19]. In the Jimbee genotype, the renal localization and abundance of GLUT1, GLUT2, and SGLT1 were not changed [56]. Histomorphometric analysis of the kidneys of Jimbee mice revealed that the renal cortex had a normal structure and morphology, but shrinkage was observed in the glomerular and tubular apparatus [56]. Moreover, immunofluorescence analysis of renal sections revealed the absence of SGLT2 from the apical membrane of the proximal infusion tubules, but positive vesicles remaining in the cytoplasm or at the perinuclear interface were detected [56]. The Jimbee mouse may be a useful model for studying the effects of loss of SGLT2 function on kidney structure and physiology.

### 3.4. SAMP10-ΔSglt2 Mouse

SAMP10-ΔSglt2, a SGLT2 spontaneous mutant mouse line, was established in the SAMP10 mouse, a model of accelerated aging [20]. The senescence-accelerated mouse (SAM) was developed by a group at Kyoto University [57]. An inbred senescence-accelerant mice (SAMP) line and the senescence-resistant mouse model (SAMR) line were reported in 1981 [58]. SAMP and SAMR mice exhibit no glucosuria.

SGLT2 mRNA expression was reduced to 34% in SAMP10-ΔSglt2 compared to SAMR1 mice. No differences were observed for the other renal glucose transporters SGLT1, GLUT1, and GLUT2 [20]. The mutation site of SGLT2 is a loss of guanine at position 1236. This results in a frameshift mutation with a stop codon at position 1264, producing a truncated protein of 421 amino acids. A protein truncated from the eighth transmembrane domain was deduced [20]. SAMP10 mice are characterized by accelerated aging, depressive-like behavior, and brain atrophy [59]. Glycosuria-prone line (SAMP10-ΔSglt2) mice weighed less than mutation-free line (SAMP10(+)) mice until 8 months of age [60]. SAMP10-ΔSglt2 consumed more food than SAMP10(+) [60]. The blood glucose levels of SAMP10-ΔSglt2 were significantly lower than those of SAMP10(+) and SAMR1 at 4 and 6 months of age. Age-related brain atrophy was not affected by the SGLT2 mutation [60]. In SAMP10-ΔSglt2, there was no effect of aging on urinary glucose excretion [20]. Learning and memory abilities were examined in a passive avoidance test. Although there were no significant differences in acquisition trials, SAMP10-ΔSglt2 mice had significantly shorter memory retention than SAMP10(+) and SAMR1 in a test conducted 24 h after acquisition trials at 12 months of age. In other words, SAMP10 with mutations in SGLT2 exhibited decreased memory capacity with aging.

#### Amyloid Precursor Protein (APP) and Amyloid Precursor-like Protein (Aplp) in the Hippocampus of SAMP10-ΔSglr2

SAMP10-ΔSglt2 and SAMP10(+) had the same genetic background, but SAMP10-ΔSglt2 exhibited a more pronounced decline in memory retention with aging. At 2 months of age, before morphological changes had been observed, a transcriptome analysis of SAMP10-ΔSglt2 and SAMP10(+) hippocampi was performed [60]. The expression of *Aplp1* was significantly downregulated in SAMP10-ΔSglt2 compared to SAMP10(+). Therefore, changes in hippocampal expression of *Aplp1*, *APP*, and *Aplp2* were compared for SAMP10-ΔSglt2 and SAMP10(+) at 2 and 11–12 months of age by quantitative real-time reverse transcription PCR (qRT-PCR). The comparison of *APP* expression in the hippocampus at 11–12 months of age showed significantly increased expression in SAMP10-ΔSglt2 compared to SAMR1 (Figure 1). As APP is the source of amyloid-β (Aβ) peptides, the marked increase in APP expression with aging may contribute to the memory loss in SAMP10-ΔSglt2. In contrast, *Aplp1* was markedly downregulated in the hippocampus at 2 months of age [60]. Lower *Aplp1* levels at younger ages may be more important for age-related cognitive decline than Aβ accumulation. The expression of *Aplp2* decreased with aging in these lines of mice, and there was no difference between them (Figure 1).

## 4. Discussion

To explore the role of SGLT2 in kidney and glucose metabolism and to study the effects of long-term use of SGLT2 inhibitors, experimental animals with mutations in SGLT2 causing renal glycosuria were developed. Four cases are discussed herein. Despite the presence of SGLT2 mutations, three of these mutant mice did not exhibit significant reductions in blood glucose levels, similar to many cases of renal glycosuria in humans.

SGLT2^−/−^ mice with a db/db genetic background exhibited a dramatic improvement in hyperglycemia and plasma insulin levels as well as preserved pancreatic beta cell function, indicating the benefit of SGLT2 mutant mice as a means of demonstrating the efficacy of SGLT2 inhibition. When low doses of streptozotocin were administered to SGLT2^−/−^ and Sweet Pee mice to create a diabetes model via impaired insulin secretion, hyperglycemia was improved in both cases, but mortality was reported to be increased in Sweet Pee compared to wild-type mice. Mutations in SGLT2 may increase the risk of malnutrition and mortality. This indicates a cautionary point in the use of SGLT2 inhibitors.

Hyperglycemia impairs the differentiation of bone marrow mesenchymal cells into adipocytes, while osteokines, a bone-derived factor, may regulate glucose homeostasis [63]. Hypercalciuria is a well-known phenomenon in children with genetic SGLT2 deficiency [12]. Poor glycemic control correlates with excessive urinary calcium loss and subsequently stimulates chronic parathyroid hormone (PTH) secretion, which adversely affects the skeleton. Improved glucose control is associated with the normalization of urinary calcium excretion, which averts the stimulation of PTH secretion and may have a positive impact on bone density [63]. Thus, the crosstalk between bone and glucose homeostasis has been experimentally confirmed.

In Sweet Pee and Jimbee mice, effects on bone metabolism have been suggested. Type 2 diabetes is a risk factor for osteoporosis. The effects of diabetes medications on bone and mineral metabolism have been observed. SGLT2 inhibitors may not increase the risk of bone fracture, but their effects are controversial and require further study [64]. SGLT2 inhibitors are also thought to promote renal phosphate reabsorption, increase fibroblast growth factor 23 and PTH, and act by decreasing active vitamin D, causing calciuria [65]; however, the effects of diabetes medications on bone health need further study. Information on changes in bone and mineral metabolism in SGLT2 mutant mice is vital in order to effectively explore the effects of SGLT2 inhibitors.

The brain requires large amounts of glucose. Although physiological blood glucose levels are tightly controlled by endocrine responses, rapid changes in blood glucose, such as hypoglycemia or hyperglycemia in intra-day and inter-day blood glucose variability, can occur. Hyperglycemia produces free radicals and impairs the endogenous antioxidant defense system [39]. Abnormal metabolism of glucose leads to a glycation reaction, producing advanced glycation end products. This is believed to be the key link between diabetes and Alzheimer’s disease. In addition, both hypoglycemia and hyperglycemia have been demonstrated to play an important role in the development of Alzheimer’s disease [66]. Hypoglycemia may consequently lead to the loss of one’s nerves and also contribute to diabetic vascular disease via oxidative stress [39]. In diabetic patients, hypoglycemia often occurs due to high doses of insulin. Severe hypoglycemia has been associated with cognitive decline, including episodic memory, in elderly patients with type 1 diabetes [67]. In young children with type 1 diabetes, not only hypoglycemia but also hyperglycemia can impact cognition and brain development [68]. Severe hypoglycemia induces neuronal damage in vulnerable areas, such as the hippocampus and cerebral cortex, and cognitive deficits associated with hypoglycemic coma have been reported to be associated with neuronal death in the hippocampus. Although the mechanism is not fully understood, many factors have been identified, including excitotoxicity, oxidative stress, and mitochondrial dysfunction [69].

As a novel function of the APP family, expression of APP and Aplp2 has been reported to regulate plasma insulin, glucose concentration, and body weight [40]. Since a significant weight loss of approximately 10% was observed in Aplp1 knockout mice compared to controls [70], it is possible that Aplp1 is also involved in glucose metabolism as a member of the APP family. Growing evidence suggests that adult hippocampal neurogenesis plays an essential role in learning and memory. All members of the APP family are involved in synaptogenesis and synaptic plasticity, but Aplp1 is particularly essential for synaptic maintenance and can increase neurogenesis [71]. It is also supposed that only Aplp1 evolved to acquire a synaptic adhesion property [72]. Despite the suggested importance of Aplp1 in the brain, few significant changes have been observed in Aplp1 knockout mice due to the compensatory effects of members of the APP gene family. However, it was reported that loss of Aplp1 increases excitatory neurotransmission and decreases network inhibition [73], and that hippocampal neurons lacking APP family members exhibit hyperexcitability [74]. APP was also shown to regulate hippocampal neurogenesis by maintaining the balance between excitation and inhibition in GABAergic interneurons [75].

Glucose is taken up by glucose transporters: GLUT1 is mainly localized in the luminal and abluminal membranes of endothelial cells in the blood-brain barrier, and GLUT3 is highly expressed in neurons [76]. Recently, the Aβ peptide was reported to interfere with the membrane localization of GLUT1 in astrocytes, resulting in decreased glucose metabolism in the brain [43]. An important function of astrocytes is the supply of glucose to neurons. These results suggest that dysregulation of blood glucose regulation in the brain parenchyma may be a central phenomenon in Alzheimer’s disease.

SAMP10-ΔSglt2, a mutant of the senescence-accelerated mouse model (SAMP10), exhibited mild long-term hypoglycemia and age-related memory loss. *Aplp1* was significantly downregulated in the 2-month-old hippocampus [60]. Since Aplp1 is particularly essential for synaptic maintenance, a decline in Aplp1 levels at younger age may be important for age-related cognitive decline. Another factors that was found to contribute to age-related memory loss is the increased expression of APP in brains of older age. An increase in Aβ with an increase in APP is proposed to interfere with membrane translocation of GLUT1 in the brain [43], resulting in a hypoglycemic state in the brain. This may also reduce hippocampal neurogenesis and impair brain function. It is highly expected to clarify the mechanism of changes in APP and Aplp1 expression associated with SGLT2 mutation.

Similar to SAMP10, SAMP8 is a member of the SAMP lineage that exhibits accelerated aging. The characteristics of SAMP8 are to exhibit decreased memory from a relatively early age and to show accumulation of Aβ in the brain, making it a possible model for Alzheimer’s disease [77,78,79]. SAMP8 does not exhibit urinary glucose but has been reported to have decreased glucose uptake in the brain [80]. The diabetic drug metformin has also been reported to improve learning and memory performance by reducing the accumulation of Aβ in the brain of SAMP8 [81]. These findings may suggest that APP accumulation results in hypoglycemia in the brain, leading to impaired brain function.

It is not clear as to why the SGLT2 mutation results in altered expressions of *APP* and *Aplp1*, but because the APP family is thought to be involved in glucose metabolism, it is speculated that the changes in blood glucose levels associated with SGLT2 mutations are important. The increased *APP* expression associated with SGLT2 mutations may provide a good model with which to investigate the relationship between glucose metabolism and Alzheimer’s disease.

## 5. Conclusions

Four lines of SGLT2 mutant mice established so far have yielded the following results.

SGLT2 mutant mice are valuable as a means of demonstrating the efficacy of SGLT2 inhibition.SGLT mutation and inhibition may increase the risk of malfunction.The mutant mice are important for information on changes in bone and mineral metabolism due to SGLT2 inhibition.The mutant mouse is an interesting model for investigating the relationship between glucose metabolism in the brain and Alzheimer’s disease.

In addition to the research using established SGLT2 mutant mice, the development of new mutant mice is also expected in order to understand the functions of SGLT2, not just sugar reabsorption.

## Figures and Tables

**Figure 1 ijms-24-06278-f001:**
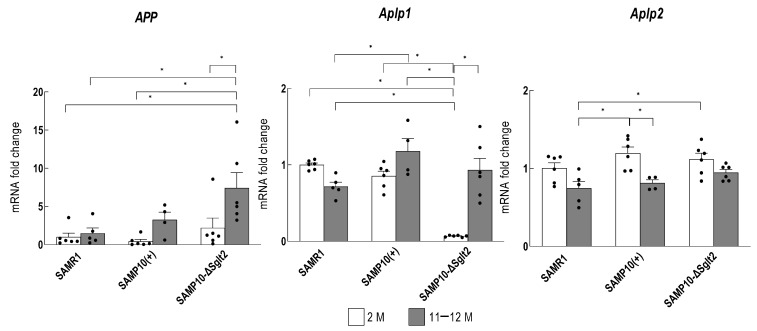
Expression of *APP*, *Aplp1*, and *Aplp2* in hippocampi of male SAMP10-ΔSglt2, SAMP10(+), and SAMR1. The levels were measured at 2 and 11–12 months of age (*n* = 4–6, * *p* < 0.05). qRT-PCR analysis was performed using Power Up SYBR Green Master Mix (A25742, Applied Biosystems Japan Ltd., Tokyo, Japan) and an automated sequence detection system (Step One, Applied Biosystems). The relative expression of the *APP* and *Aplp2* gene was determined using the previously validated primers [61,62]. β-actin was used as an internal control. Data for *Aplp1* were taken from the results in [62]. Data are expressed as means ± SEM overlaid on scatter plots. Black dots indicate values for each mouse. The data were analyzed by ANOVA followed by the Tukey–Kramer method.

## Data Availability

Not applicable.

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
