# Peer review of "Mouse Models with SGLT2 Mutations: Toward Understanding the Role of SGLT2 beyond Glucose Reabsorption"

_ijms, 2023, doi:10.3390/ijms24076278_

Round 1

Reviewer 1 Report

This interesting review describes the different mutations of SGLT2 and the link between SGLT2 mutation and Alzheimer's disease. The paper in clear, innovative and well-written.

My comments: Alzheimer's disease must also be cited in the introduction.

I also suggest improving the introduction on SGLT2i as in the reference (https://pubmed.ncbi.nlm.nih.gov/34959126/)

Author Response

Comments and Suggestions for Authors

This interesting review describes the different mutations of SGLT2 and the link between SGLT2 mutation and Alzheimer's disease. The paper in clear, innovative and well-written.

My comments: Alzheimer's disease must also be cited in the introduction.

I also suggest improving the introduction on SGLT2i as in the reference (https://pubmed.ncbi.nlm.nih.gov/34959126/)

Thank you very much for reviewing our manuscript.

We added the explanation about Alzheimer's disease in the introduction and improved the introduction on SGLTi according to the reviewer’s comment.

In addition, following suggestions from other reviewers, I rewrote the entire introduction and abstract, and rewrote the article with completely changed structure and chapters.

Reviewer 2 Report

1. The conclusions are very long and are more typical of the discussion due to the data provided. Please try to summarize in 1-2 short paragraphs the most relevant conclusions of the work.

2. Figure 2 and the explanation, better in the discussion section

Author Response

Comments and Suggestions for Authors

  1. The conclusions are very long and are more typical of the discussion due to the data provided. Please try to summarize in 1-2 short paragraphs the most relevant conclusions of the work.

Thank you for your precise comment.

According to your suggestion, we summarized it in 1-2 short paragraphs (line 370-377).

In addition, following suggestions from other reviewers, I rewrote the entire introduction and abstract, and rewrote the article with completely changed structure and chapters.

  1. Figure 2 and the explanation, better in the discussion section.

It was still insufficient to show Figure 2, so it has been removed.

Reviewer 3 Report

The authors have set themselves an exciting task that promises a good overview of the overall effectiveness of SGLT-2 inhibitors. However, they got lost in the pile of data. The title and the abstract already lead us in a different direction than the article. The article's transparency and the narrative's red thread must recover. The story does not flow and is just a bunch of stringed data.

Suppose the authors decide to correct the article. In that case, I advise rewriting the abstract, completely changing the structure and chapters of the article, and correcting the entire red thread and fluidity of the narration. Many data need to be more relevant and not repeated.

The opening part of 2, 2.1 is written too generally, and then bone metabolism is mentioned, which no longer functions in the rest of the story. Chapters 3, 4 and part of chapter 5 can be combined and shortened. It should also mention heart failure to emphasize all the benefits of SGLT-2 inhibitors. I also need help finding the point of adding the two figures.

Overall, a shorter introduction can be written, summarizing all the effects of SGLT-2 inhibitors (Chapters 1-4) and then chapter 5, which is the article's primary goal.

Author Response

The authors have set themselves an exciting task that promises a good overview of the overall effectiveness of SGLT-2 inhibitors. However, they got lost in the pile of data. The title and the abstract already lead us in a different direction than the article. The article's transparency and the narrative's red thread must recover. The story does not flow and is just a bunch of stringed data.

Suppose the authors decide to correct the article. In that case, I advise rewriting the abstract,

completely changing the structure and chapters of the article, and correcting the entire red thread and fluidity of the narration. Many data need to be more relevant and not repeated.

Thank you for your thoughtful and precise suggestions.

We rewrote the abstract, completely changing the structure and chapters of our article.

Some new references were added (yellow colored).

The opening part of 2, 2.1 is written too generally, and then bone metabolism is mentioned, which no longer functions in the rest of the story. Chapters 3, 4 and part of chapter 5 can be combined and shortened. It should also mention heart failure to emphasize all the benefits of SGLT-2 inhibitors. I also need help finding the point of adding the two figures.

The explanation about Alzheimer's disease and amyloid precursor protein was added in the introduction to find the point of adding Figure 1 (line 80-86).

Figure 2 was removed.

Overall, a shorter introduction can be written, summarizing all the effects of SGLT-2 inhibitors (Chapters 1-4) and then chapter 5, which is the article's primary goal.

We revised our entire manuscript with a longer introduction and summarized chapters 2-4.

Round 2

Reviewer 3 Report

The article is now nicely written and reviewed. I noticed two typos:  Line 19 - hyperglycemia and Line 71  - weight. Thank you for following the tips. The article can be published.